# Synthesis and Properties of Octet NiCr Alloy Lattices Obtained by the Pack Cementation Process

**Peng Zhao [1,2], Deqing Huang [2,3], Hongmei Zhang [1,*], Weiwei Chen [1,*] and Yongfu Zhang [1]**

1    School of Materials Science and Engineering, Beijing Institute of Technology, Beijing 100081, China
2    Gaona Aero Material Co., Ltd., Beijing 100081, China
3    Beijing Key Laboratory of Advanced High Temperature Materials, Central Iron & Steel Research Institute, Beijing 100081, China
*    Correspondence: zhanghm@bit.edu.cn (H.Z.); wwchen@bit.edu.cn (W.C.);
     Tel.: +86-10-68912709 (ext. 109) (W.C.)

**Abstract:** NiCr alloys with different components were obtained by pack chromation and homogenization heat treatment of octet Ni lattice. The microstructure, alloy composition, microhardness and quasi-static compression properties of the NiCr lattice were tested. The results showed that after homogenization heat treatment, the NiCr alloy lattice had an austenitic structure with uniform composition. Compared with the pure nickel lattice, the microhardness, compressive strength, elastic modulus and energy absorption of the NiCr lattice increased with the increase of chromium content. The microhardness, specific strength, specific modulus and specific energy absorption of the Ni-45Cr alloy were 363 HV, 11.1 MP/(g/cm$^3$), 1169.1 MP/(g/cm$^3$) and 10 J/g, respectively, which were attributed to the solid solution strengthening provided by chromium and the increase in density. NiCr alloy lattices have high strength and toughness and may have potential applications in high-temperature filters or heat exchangers.

**Keywords:** mechanical properties; nickel-based lattice; pack cementation; solution strengthening

## 1. Introduction

Metal lattice is a new type of structural material with rapid development in current materials science. Compared with porous materials such as disordered foams and sponges, lattice materials are ultralight and have high specific strength, specific stiffness and good energy absorption capacity [1–5]. In recent years, research on Ni-based microlattice materials has mainly focused on pure Ni, using the process of electrodeposition. These Ni lattices have good mechanical properties [6–9]. However, compared to Ni-based alloys, porous Ni prepared by the electrodeposition process has poor oxidation resistance and corrosion resistance [10–12]. Dunand found that after 50 h of oxidation at 1000 °C, compared to a pure Ni foam, the weight gain rate of the Ni–19.4%Cr foam was reduced two times and that of the Ni–27.2%Cr foam by a factor of four. Therefore, it is necessary to develop nickel-based lattice materials with a high melting point, oxidation resistance and excellent mechanical properties [10].

Nickel has a special electronic structure, and alloying elements can be added through pack cementation technology [10,13], electrophoretic co-deposition [14], beam vapor deposition and other process to improve the room-temperature [15–17] and high-temperature mechanical properties and the heat resistance of nickel-based alloys. NiCr, NiCrAl and NiCrFe alloy foams reported by Dunand [10,18–20], Chyrkin [17] and Qiu Pang [11,13,21–23] had excellent oxidation resistance and good mechanical properties. Dunand prepared a NiCr alloy foam with different Cr contents, such as Ni-12Cr, Ni-22Cr and Ni-27Cr, by pack cementation and found that with the increase of Cr content, the relative density of the metal foam increased from 2.2% to 3.2%, and the strength increased from 0.11 Mpa to 0.9 MPa [10]. John H. Martin obtained NiCuAlTi Monel alloys and NiCrAl superalloys

for hollow Ni lattice-embedded alloy elements, which further proved the feasibility of preparing Ni-based superalloy lattices by solid-powder embedding [24]. Erdeniz coated Cr, Al and Ti elements on the surface of a pure Ni microlattice via pack cementation and achieved, after homogenization and aging treatment, a uniform Ni-Cr-Al-Ti composition with a $\gamma/\gamma'$ superalloy microstructure. The compression properties of the lattices with different components were measured in air at 788 °C. After normalizing for density ($\sigma/\rho$), the specific strengths of single-phase and $\gamma'$ strengthened Ni-(10–14)Cr-3Ti-1Al (wt%) lattices reached 8.1 MPa/(g/cm$^3$) and 4.8 MPa/(g/cm$^3$). Compared to pure nickel lattices, they were about 3.7 times and 2.2 times higher, respectively. The high high-temperature compressive strength of Ni-Cr-Al-Ti resulted from the solution strengthening provided by Cr, Al and Ti (as well as from the fine $\gamma'$ precipitation that may be formed in aging), along with enhanced oxidation resistance provided by Cr and Al [25]. The octahedral lattice prepared by this method had high specific strength, but the actual strength was relatively low, and it could not withstand large loads due to its low relative density (less than 2%).

In the study of multiple lattice materials, the octet structure has received extensive attention. The octet lattice consists of 36 rods of exactly the same length and diameter. Eight regular tetrahedrons are distributed on the surface surrounding the central regular octahedron (Figure 1). The Maxwell's criterion defines M = b − 3j + 6, b, where j represents the number of rods and nodes in the cell. If M ≥ 0, the lattice structure is stretch-dominated, otherwise it is bend-dominated. Ideally, the relative modulus (relative strength) and relative density of the stretch-dominated structure satisfies $E/E_s \propto \rho/\rho_s$, $\sigma/\sigma_s \propto \rho/\rho_s$, whereas the bend-dominated structure satisfies $E/E_s \propto (\rho/\rho_s)^2$, $\sigma/\sigma_s \propto (\rho/\rho_s)^{1.5}$. E, $\sigma$ and $\rho$ represent the elastic modulus, yield strength and density of porous materials, and Es, $\sigma_s$, $\rho_s$ represent the elastic modulus, yield strength and density of solid materials. The closer the value of the power exponent is to 1, the slower the decrease in intensity and modulus as the density decreases. The structure of a nickel foam is disordered, similar to the Kelvin model, which is a curved-dominated porous material, while the octet lattice is a stretch-dominated material. According to Maxwell's theory and combined with the actual properties of the octahedral Ni lattice, when the material and density are the same, the NiCr alloy with an octet structure will have better mechanical properties and mass efficiency than a bend-dominated lattice and a Ni-based foam [7,9,26,27].

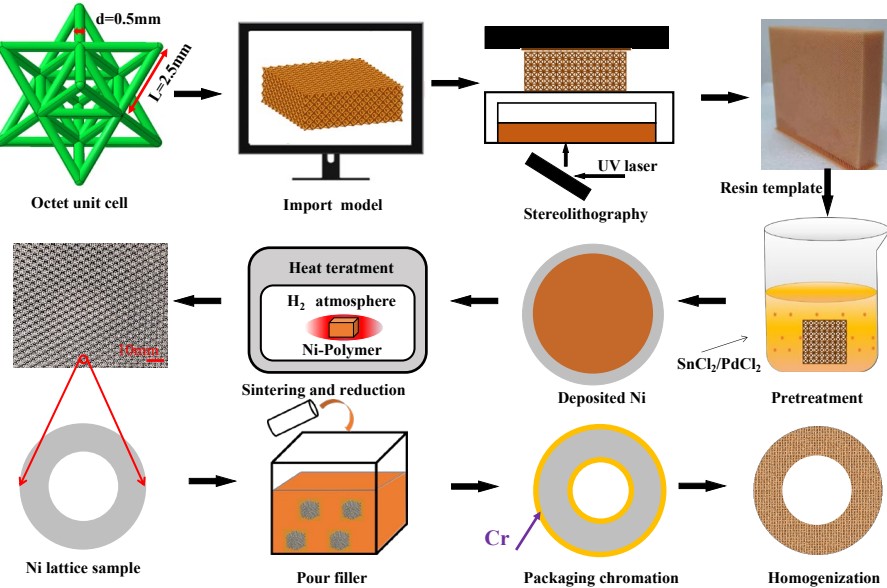

**Figure 1.** Schematic diagram of the preparation process of NiCr alloy lattices.

In this paper, an electrodeposited Ni octet structure lattice was used as a template, and a nickel–chromium alloy lattice with high strength and toughness was prepared by pack cementation. Among porous materials, the nickel–chromium lattice has great advantages

because of its the mechanical properties. It is thus a candidate for the productions of, e.g., as high-temperature catalyst media, high-temperature filters or heat exchangers, and porous material carriers.

## 2. Materials and Methods

By combining the preparation processes of a NiCr foam and a NiCrAlTi microlattice, a three-dimensional model of the octet lattice was established firstly, to obtain a rod with a diameter of 0.5 mm and a length of 2.5 mm (Figure 1). The model was imported into a 3D printer, and the polymer template was prepared by stereolithography. Hydroxides, potassium permanganate and chromium trioxide solutions were used for surface coarseization. Then, by immersing the polymer lattice in a solution containing stannous chloride and palladium chloride, a layer of catalytic metal particle palladium was adsorbed on the surface of the resin, forming the nucleation centers. The sample was first electroless-Ni plated and electrodeposited, and then the surface coating was removed to expose the polymer skeleton to the open air [6–9,28]. Then, the lattice with the polymer was sintered at 600 °C to remove the polymer skeleton, and the hollow Ni lattice was reduced under a hydrogen atmosphere at 900 °C.

The complete pack cementation process mainly included two steps: packaging chromation and homogenization heat treatment. The packaging chromation process allowed mixing the aluminum, Cr powder (source) and ammonium chloride (activator) as fillers according to the mass ratio of 65:30:5. After about 20 min of mechanical mixing, the filler was poured into a high-chromium stainless steel shell, and the nickel lattice was inserted. The housing was placed at the water-cooled end of a tube furnace and pushed into the hot zone of the furnace at a processing temperature of 1050 °C (RYJ-2000C, Gaona Aero Material Co., Ltd., Beijing, China) for different times, depending on the desired composition. The packet was then quickly pushed into the other water end of the furnace and cooled to ambient temperature. After this first heat treatment step, the alloyed lattice was homogenized at 1200 °C for 48 h under flowing Ar, and the Cr-rich coating on the inner and outer surfaces of the Ni hollow truss was diffused into the walls [10,11,18–25].

To compare the mechanical properties of NiCr and NiCrFe alloy foams, we used a Cr content of the alloy lattice basically equivalent to that of the NiCr foam prepared by Dunand and Qiu Pang et al. [10,11,20]. We then increased the Cr content as much as possible (up to 45%) to obtain higher strength and modulus. Scanning electron microscopy (SEM) was used to observe the surface topography of the NiCr lattice, and EDS was used to measure the elemental distribution of the hollow rod cross section. The metallographic structure of the Ni and NiCr lattices was observed by optical microscopy, and the Feret diameter distribution of 100 grains was extracted from the image by using Image J software. X-ray diffraction (XRD) was used to analyze the phase of the lattice specimens. Their microstructure was observed by using transmission electron microscopy (TEM). A QNESS Q30A+ microhardness tester was used to measure the microhardness of lattice specimens of different compositions. The NiCr lattice material was embedded in epoxy resin, its cross-section was ground and polished. A pressing load of 100 g (quality) was applied to the rods section with a loading time of 10 s. The average measurement of 10 arbitrary points was specified as the microhardness of the sample. Finally, the quasi-static compression performance of the NiCr hollow nickel lattice materials with different compositions were tested by a universal material testing machine. All the compressed samples contained $4 \times 4 \times 4$ cells, and the indenter displacement speed was set to 2 mm/min.

## 3. Results and Discussion

### 3.1. Microstructrue Characterization

Figure 2a shows the local topography of a hollow nickel lattice sample after annealing. As can be seen in the figure, a periodic hollow tube array was obtained by sintering removal of the polymer and a high-temperature reduction treatment, and its basic unit was the octet structure. The hollow nickel lattice surface was highly dense, and there were no obvious

holes and cracks. As can be seen in Figure 2a,d, the Ni lattice had straight grain boundaries after heat treatment, and the shape of the grains was regularly polygonal. Figure 2c shows the size distribution of pure Ni lattice grains, with an average particle size of 11 μm. The grain size was mainly concentrated in the range of 5~15 μm, which accounted for 80% of the total number of grains.

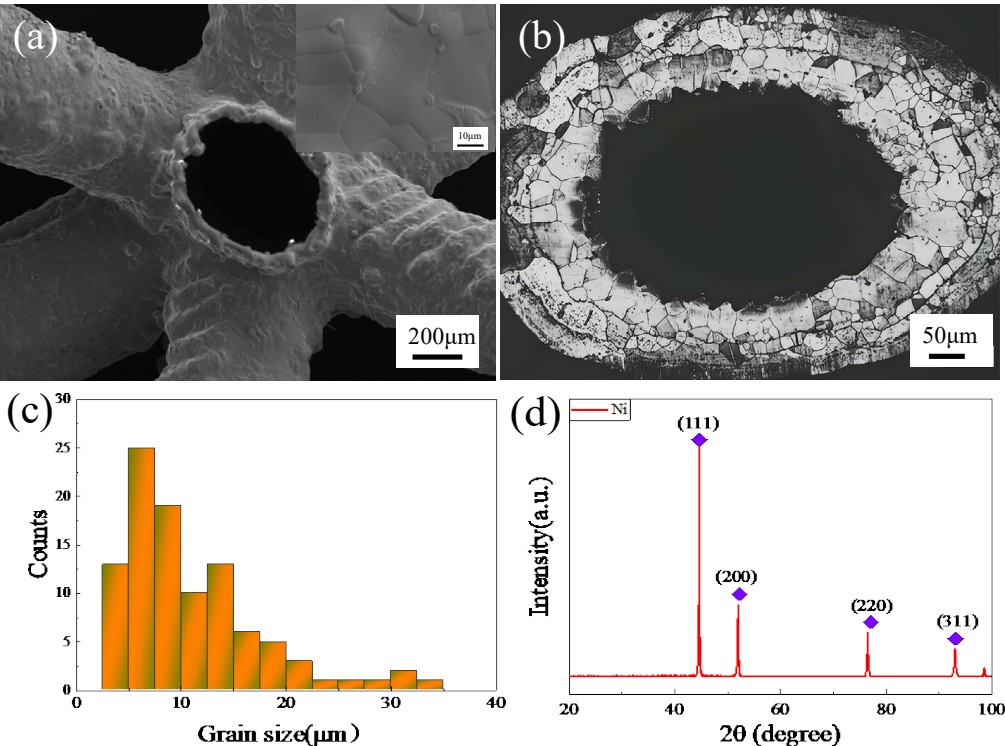

**Figure 2.** Characterizing the basic properties of pure nickel lattices: (**a**) SEM micrographs; (**b**) optical micrographs of the etched cross section of a lattice strut; (**c**) grain size distribution; (**d**) XRD diffraction spectrum.

Figure 2d shows the XRD patterns of Ni lattices. It can be seen that there were significant diffraction peaks at 44.51°, 51.9° and 76°. They corresponded to the (111), (200) and (220) crystal faces of Ni and were consistent with the nickel powder peaks on the diffraction standard card JCPDS: 04-085.

The surface morphology of NiCr alloy lattices with different compositions was basically similar. Figure 3a shows the macroscopic morphology of the NiCr lattice, Figure 3d presents the outer surface morphology of the hollow rod and Figure 3b,e the morphology of the inner surface of the rod. As can be seen in the Figures, the hollow rod cross sections were circular and elliptical, and there were no obvious distortion and deformation in the lattice rod. The NiCr alloy lattice maintained the original macro-structure of the octet hollow Ni lattice. This showed that the Ni lattice was not seriously damaged by the high-temperature heat treatment process, and the NiCr alloy still had the excellent structural properties of the octet. Compared with the initial Ni lattice surface topography (Figure 2a), the surface roughness of the alloy lattice increased, which could be due to vapor deposition on both the inner and the outer surfaces during the packaging chromation and vapor-phase diffusion coating processes. The roughness of the outer surface was greater than that of the lattice rod, which could be caused by the combination of electrodeposition and packaging chromation. The few white bright spots on the rod surface in the Figure were $Al_2O_3$ particles (used as the filler powder). They were attached to the inner and outer surfaces of the NiCr lattice during the packaging chromation process [11].

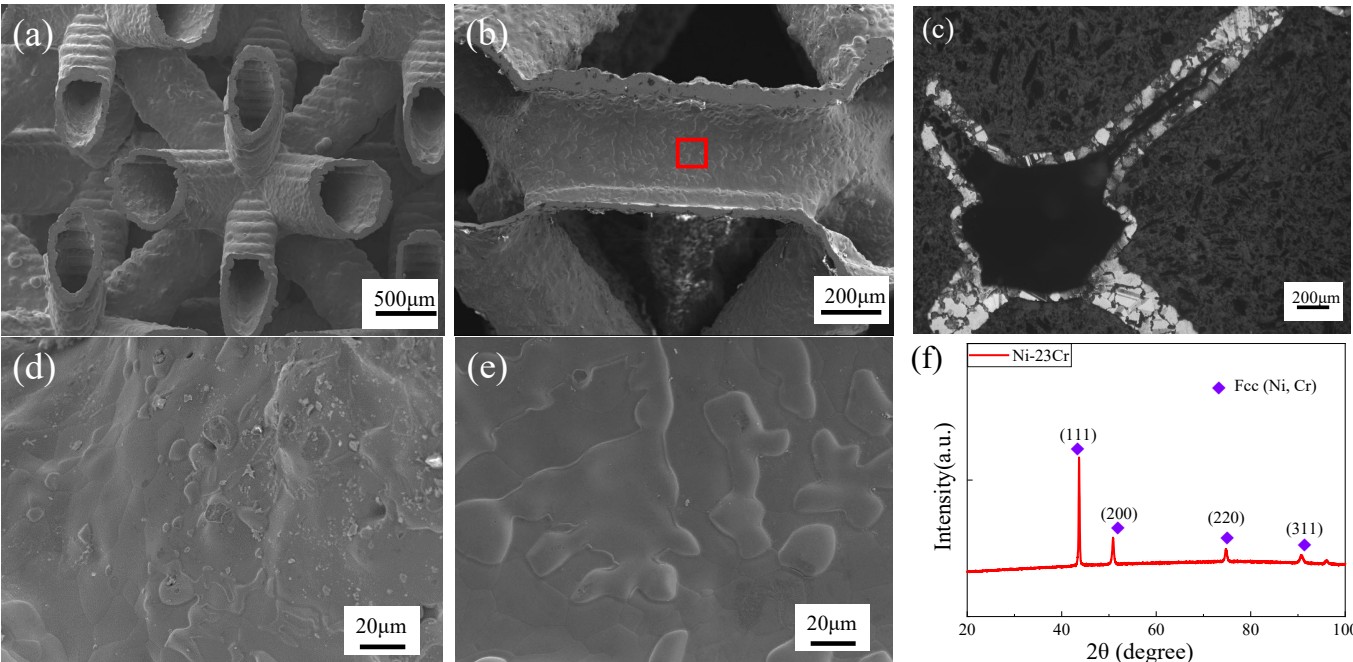

**Figure 3.** SEM micrographs of (**a**,**d**) the outer surface of the NiCr lattice and (**b**,**e**) the inner surface of the NiCr lattice; (**c**) optical micrographs of the etched cross section of a NiCr lattice strut; (**f**) XRD diffraction spectrum.

Figure 3c shows the typical metallographic structure of the Ni-23Cr lattice material after homogenization. It can be seen that there was no obvious alloy layer in the NiCr alloy structure after homogenization. The microstructure of NiCr was a homogeneous (Ni, Cr) austenite structure with distinct grain boundaries [10,25]. Compared with the grain size of the heat-treated Ni lattice, the grains of the lattice alloy had grown after homogenization, and the size was about 40~80 μm, so that 1–2 grains could span the entire lattice wall.

Figure 3f shows the XRD diffraction spectrum of the NiCr lattice. It can be seen that the homogenized lattice NiCr alloy surface did not present a pure Ni phase or Cr phase, but only a single Ni solid-solution phase existed. We speculate that this was because the atomic radii of Ni and Cr (0.162 nm and 0.185 nm, respectively) and the lattice constants (0.353 nm and 0.289 nm, respectively) were very close. In a high-temperature environment at 1200 °C, the two elements Ni and Cr would diffuse toward each other: Cr atoms from the surface inward, and Ni atoms from the inside to the outside, to form an FCC (Ni, Cr) solid solution [11,29].

Figure 4 shows a TEM bright-field image of a NiCr alloy lattice. It can be seen that there was a large number of dislocations in the grains. The solid solution of Ni and Cr causes a lattice distortion to a certain extent, which increases the resistance of the dislocation motion, making slippage difficult and leading to an increase in the strength and hardness of the solid-solution alloy [30–32].

The packaging chromation time was set to 4 h, 8 h, 12 h and 16 h to obtain lattices composed of Ni-16Cr, Ni-23Cr, Ni-34Cr and Ni-45Cr alloys. The morphology of the rod section after heat treatment by homogenization of the crystal lattice of the four NiCr alloys and the elemental distribution of the cross sections are shown in Figure 5. The orange lines of the double arrows in Figure 5e–h represent the typical wall thickness of the lattice materials of different alloys. The red lines represent the distribution of elemental Ni, and the green lines represented the distribution elemental Cr. The yellow box in Figure 5i shows the selection region of the Ni-45Cr alloy lattice section EDS mapping test, and Figure 5j–l shows the distribution of the two elements Ni and Cr in the selection area. As shown in Figure 5, the average wall thickness $\bar{t}$ of Ni-16Cr, Ni-23Cr, Ni-34Cr and Ni-45Cr alloy lattice materials were about 25 μm, 32 μm, 38 μm and 53 μm, respectively. It can be seen in the Figure that the

lattice film was gradually thickening as the Cr content increased. In all lattice cross sections, the Ni and Cr elements were evenly distributed without significant concentration gradients. This indicated that after 48 h of homogenization, sufficient diffusion was generated between the two elements.

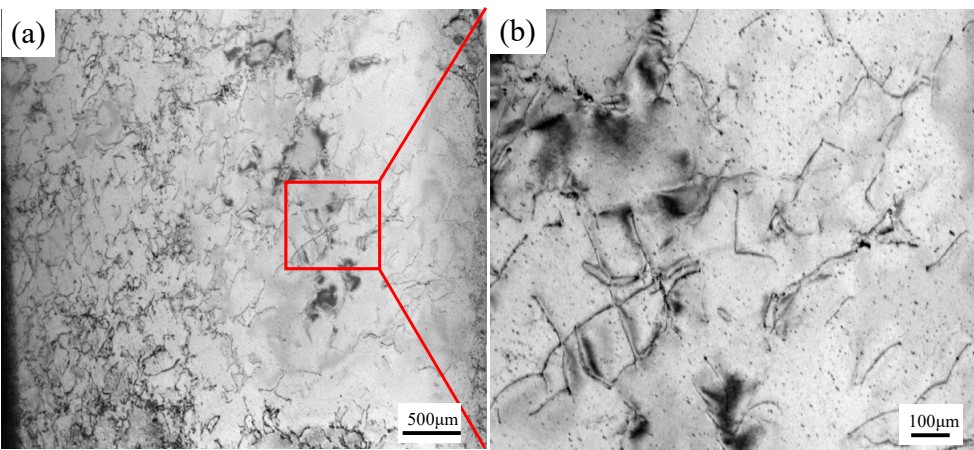

**Figure 4.** TEM microstructure of the Ni-23Cr alloy lattice: (**a**) bright-field image of the NiCr alloy lattice, (**b**) enlarged view of the red area in (**a**).

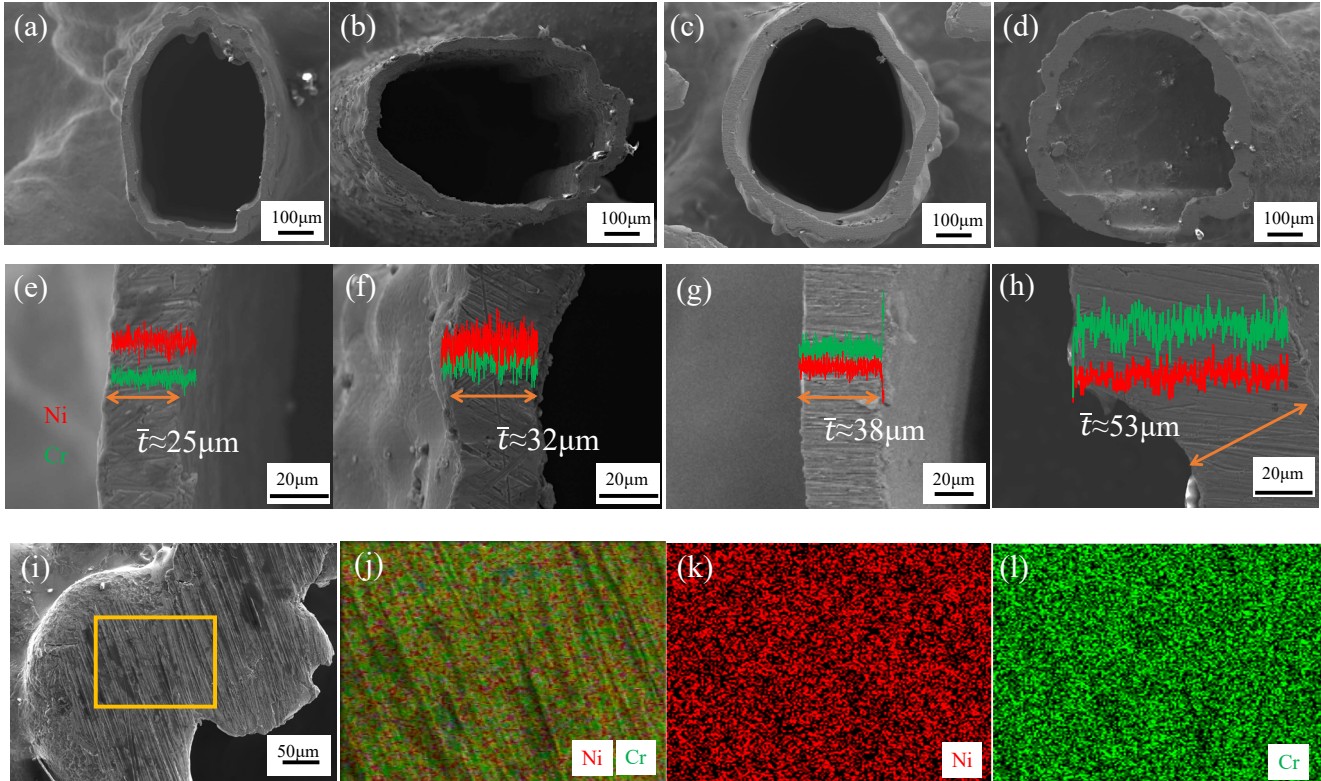

**Figure 5.** EDS results of NiCr alloy lattices with different compositions: (**a**,**e**) Ni-16Cr; (**b**,**f**) Ni-23Cr; (**c**,**g**) Ni-34Cr; (**d**,**h**) Ni-45Cr, (**i**–**l**) EDS mapping of the Ni-45Cr lattice cross section.

### 3.2. Microhardness

Figure 6 shows the microhardness of the lattices with different compositions. The average microhardness of the pure nickel lattice was 90 HV, which was roughly equal to that of the Ni foam Dunand used to make NiCr and NiCrAl alloys [10,20]. It was lower than the microhardness of an electroplated Ni lattice (295 HV) [25]. This was because

electrodeposited Ni had an ultra-fine grain structure, and the growth of the grains occurred during the annealing process at 900 °C.

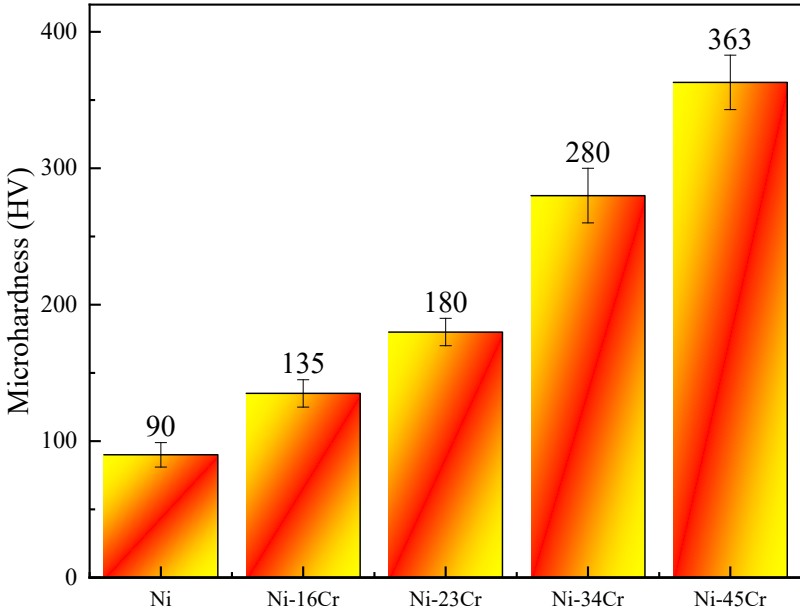

**Figure 6.** Microhardness of Ni and NiCr alloys.

The microhardness of Ni-16Cr, Ni-23Cr, Ni-34Cr and Ni-45Cr alloys was 135 HV, 180 HV, 280 HV and 363 HV, respectively. The microhardness of the NiCr alloy lattice increased with the increase of Cr content. Compared with the microhardness of the annealed Ni lattice, the microhardness of the Ni-45Cr alloy lattice was about four times higher. This was attributed to the solution strengthening provided by the Cr element, which increased with the increase of Cr content. This trend was observed for the NiCr alloy foams [10], but the microhardness of the NiCr alloy in this paper was higher.

### 3.3. Compression Curve and Mechanical Properties

Figure 7 showes the stress–strain curve corresponding to the quasi-static compression of the Ni and NiCr alloy lattices at room temperature. It can be seen in the Figure that during the quasi-static compression process, the lattices presented a linear elasticity region, a plateau region and densification stages, showing the deformation characteristics of typical elastoplastic porous materials. In the linear deformation region, the compressive stress increased linearly with the increase of the strain [33]. The slope of the initial linear phase of the stress–strain curve provided the compression modulus E of the NiCr lattice. At a strain of about 0.05, the stress reached a non-proportional stress Rp1.0, corresponding to the compressive strength. This was followed by a long stress plateau region, where the strain increased, but the stress remained largely unchanged. With the bending, buckling and plastic deformation of the truss rod, the middle position of the lattice was compacted firstly, and the stress was released. The compressive stress was transmitted to the unde-formed skeleton part through the deformed skeleton portion, and the above process was repeated, resulting in a slow rising of the stress plateau area. All curves of the nickel-based lattice materials with different compositions showed long stress plateau regions, without significant stress peaks and stress drop stages. The long stress plateau region showed good toughness, indicating that no brittle fractures occurred at the rod and node sites during the lattice deformation. In the densification stage, the stress increased rapidly with the strain due to the compression of the pores of the lattice, and the lattice at this time was equivalent to a dense metal.

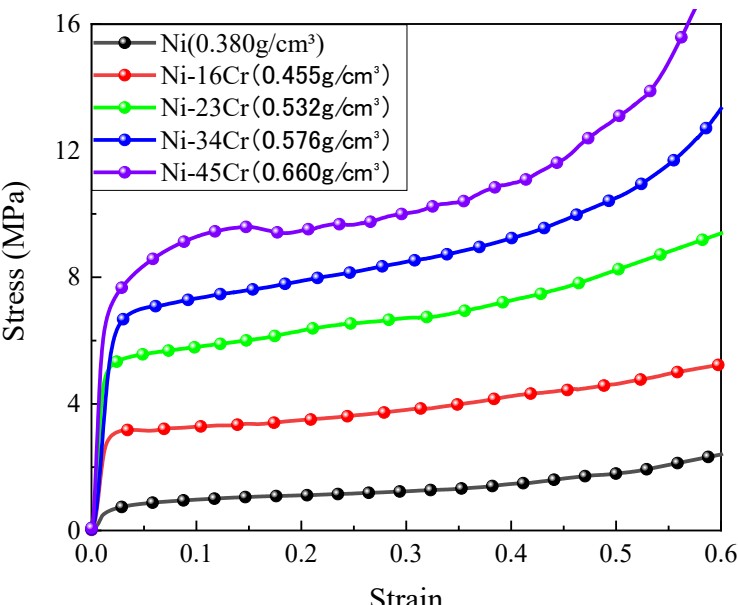

**Figure 7.** Stress–strain curves of NiCr alloy lattices with different compositions.

The mechanical properties of the lattices with different alloys are shown in Figure 8. With the increase of Cr content and density, the compressive strength and modulus of the lattice NiCr alloys gradually increased. In order to compare mechanical properties of the lattice of different alloy components together in Figure 8, the actual modulus and specific modulus needed to be multiplied by 100 on the values. (Similarly, the value of energy absorption efficiency in Figure 9 needed to be multiplied by 0.1). The compressive strength of the lattices of pure Ni, Ni-16Cr, Ni-23Cr, Ni-34Cr and Ni-45Cr alloys were 0.7 MPa, 3.1 MPa, 5.3 MPa, 6.8 MPa and 7.3 MPa, respectively. The elastic moduli were 65.2 MPa, 286 MPa, 557.7 MPa, 666 MPa and 771.6 MPa, respectively. The lattice strength and modulus of the Ni-45Cr alloy were about 10 times and 12 times higher than those of the pure nickel lattice.

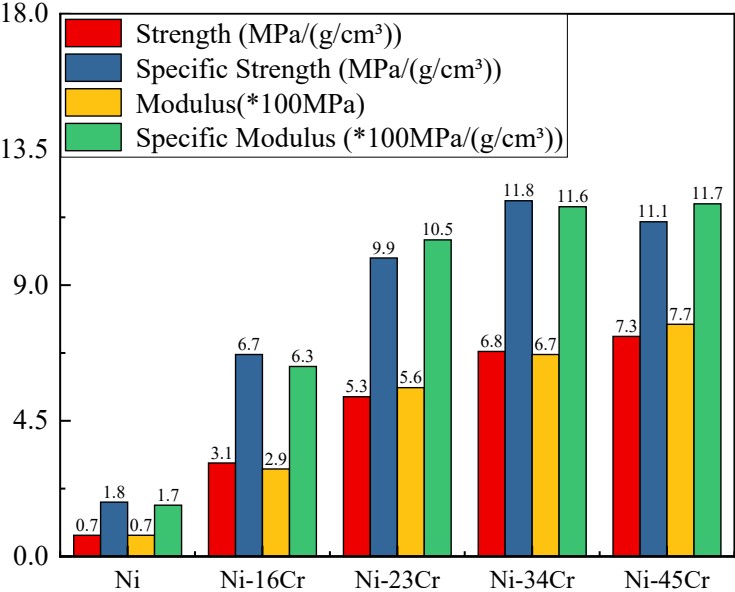

**Figure 8.** Mechanical properties of different NiCr alloys.

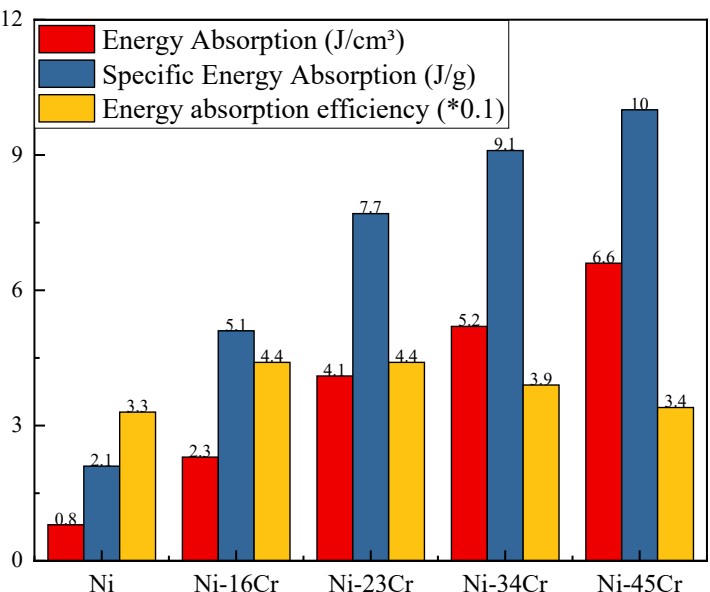

**Figure 9.** Energy absorption capacity of different NiCr alloys.

The specific strength $\sigma/\rho$ and specific modulus $E/\rho$ of a lattice can be obtained by density normalization [6,9,25]. The specific strengths of pure Ni, Ni-16Cr, Ni-23Cr, Ni-34Cr and Ni-45Cr lattices were 1.8 MPa/(g/cm$^3$), 6.7 MPa/(g/cm$^3$), 9.9 MPa/(g/cm$^3$), 11.8 MPa/(g/cm$^3$) and 11.1 MPa/(g/cm$^3$), respectively. The specific moduli were 171.6 MPa/(g/cm$^3$), 628.6 MPa/(g/cm$^3$), 1048.2 MPa/(g/cm$^3$), 1156.3 MPa/(g/cm$^3$) and 1169.1 MPa/(g/cm$^3$), respectively. The specific strength and specific modulus of the Ni-45Cr alloy were about six times and seven times higher than those of the pure nickel lattice.

The mechanical performance improvement was attributed to the increased density and the solution strengthening effect provided by the alloy elements. When the wall thickened and the bearing area became larger, the lattices were more resistant to rod buckling and lattice plastic deformation. These phenomena have been described in the study of NiCr, NiCrAl and NiCrFe alloy foams [10,20,21].

### 3.4. Energy Absorption

The energy absorption capacity wi an important technical property of porous materials. The energy absorption capacity of a hollow lattice material can be expressed by the energy absorbed per unit volume, the energy absorbed per unit of mass (i.e., specific energy absorption) and the energy absorption efficiency [34–38]. In general, the energy absorbed per unit volume W and unit of mass is calculated as:

$$W = \int_0^{\varepsilon_d} \sigma d\varepsilon \tag{1}$$

$$SAE = \frac{\int_0^{\varepsilon_d} \sigma d\varepsilon}{\rho} \tag{2}$$

To quantify the relationship between the energy absorption of different materials and the first peak stress, the energy absorption efficiency I is defined as:

$$I = \frac{\int_0^{\varepsilon_d} \sigma d\varepsilon}{\sigma_m * 100\%} \tag{3}$$

where $\varepsilon_d$ is the densification strain (the corresponding strain when the stress value reaches the peak stress value for the second time), and $\sigma_m$ generally refers to the first peak stress. However, there were no obvious stress peaks in the Ni and NiCr lattice alloy materials.

Therefore, it was specified that $\varepsilon_d = \varepsilon_a = 0.6$, and $\sigma_m$ was the stress corresponding to the compression curve when the strain was 0.6.

The long stress plateau zone made the octet lattice have a large energy absorption capacity. The external impact energy can be dissipated by the plastic work and friction between components during macroscopic deformation [39]. Figure 9 shows the energy absorption, specific energy absorption and energy absorption efficiency of Ni-based lattices with different components after homogenization. The change in energy absorption of the NiCr alloy lattice was similar to the change in compressive strength. The energy absorbed per unit volume by the pure nickel lattice was 0.8 $g/cm^3$, and the specific energy absorption was 2.1 J/g. The energy absorbed per unit volume by Ni-16Cr, Ni-23Cr, Ni-34Cr and Ni-45Cr lattice was 2.3 $J/cm^3$, 4.1 $J/cm^3$, 5.2 $J/cm^3$ and 6.6 $J/cm^3$, respectively, and the respective energy absorption values were 5.1 J/g, 7.7 J/g, 9.1 J/g and 10 J/g. The energy absorbed per unit volume and unit mass by the Ni-45Cr alloy lattice were eight times and five times higher than those of the pure nickel lattice, because the lattices showed relatively high strength after homogenization [10,21,36,38].

Octet lattice materials had high energy absorption efficiency, and the energy absorption efficiency of the octet lattice of the Ni and NiCr alloys was mainly concentrated between 30% and 40%.

### 3.5. Comparison with Other Porous Materials

The Ashby Material Properties Chart allowed us to compare multiple properties of different materials. Figure 10 shows the difference between the mechanical properties of Ni-based alloy lattice materials and other porous materials. It can be seen that the strength and modulus of homogenized NiCr lattice materials were much lower than those of solid metals and ceramics, while those foam materials had medium values.

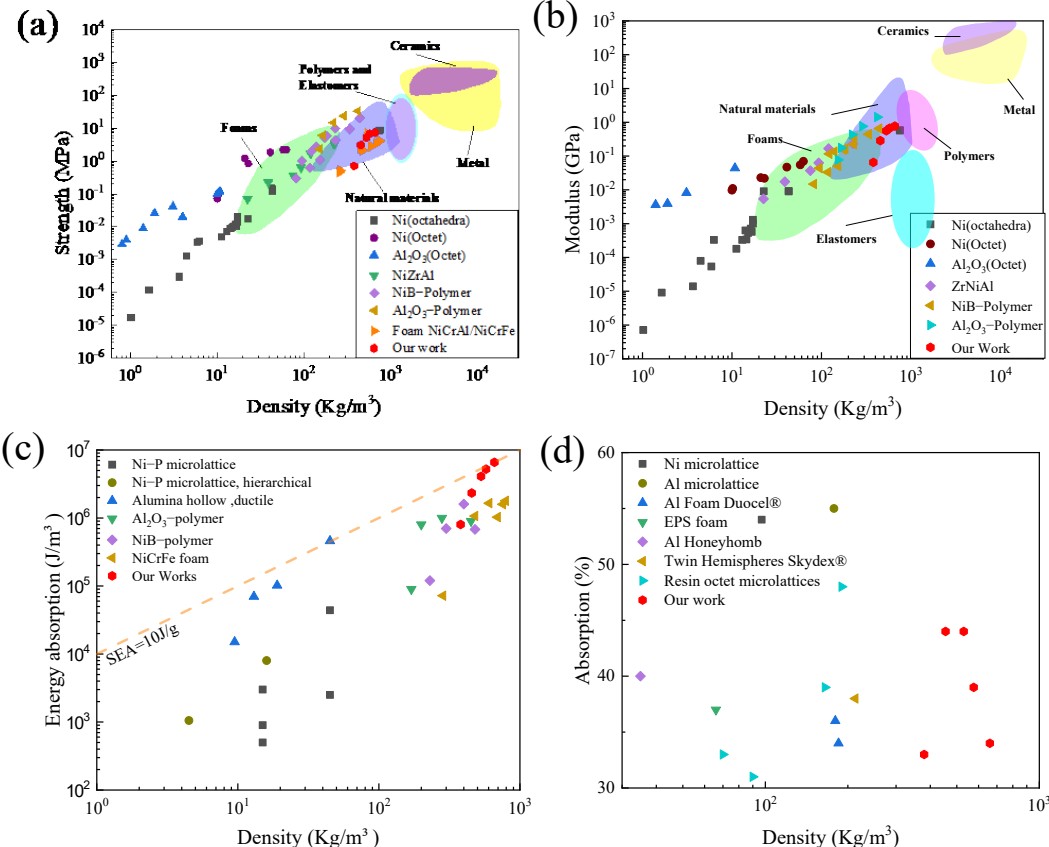

**Figure 10.** Comparison of the mechanical properties of different porous materials: (**a**) strength; (**b**) elastic modulus; (**c**) energy absorption; (**d**) energy absorption efficiency.

For a wide range of density values (between 300~700 kg/m$^3$), the mechanical properties of Ni-based alloy lattices were basically equivalent to those of the octahedral electrodeposited Ni lattice prepared by T.A. Schaedler, because of the presence of fine grains strengthening the nanocrystals in the electrodeposited octahedral Ni lattice [7]. These properties were better than those of the hollow octet NiP lattice reported by Zheng and of the ZrNiAl metallic glass lattice reported by Liontas R [9,40]. In addition, the compressive strength and elastic modulus exceeded those measured for the NiCr, NiCrAl and NiCrFe alloy foams prepared by Duand and Qiu Pang [10,11,20,21].

The octet structure Ni-based alloy lattice showed an excellent static energy absorption capacity. The energy absorbed per unit volume was higher than those of octahedral and layered NiP microlattice, Al microlattice [38], NiB-polymer lattice [41], Al$_2$O$_3$-polymer [42] lattice and NiCrFe alloy foam [17]. The energy absorption efficiency was basically equivalent to those of Al Foam Duocel$^®$, EPS foam [38] and octet resin lattices [9,35].

## 4. Conclusions

In this paper, high-strength and high-toughness NiCr alloy lattices with different compositions and densities were prepared by adjusting the pack cementation process using the octet-structure Ni lattice as the matrix. The surface morphology, cross-sectional composition and microstructure characteristics of the NiCr lattice were characterized, and the influence of the alloy composition on the mechanical properties of the alloy lattice was studied. We found that:

(1) With lattice Ni as the matrix, a hollow NiCr alloy lattice with uniform composition structure can be obtained by packaging chromation at 1050 °C and homogenization at 1200 °C for 48 h.

(2) The Cr content of the NiCr alloy lattice increased from 16% to 45%, and the microhardness increased from 120 ± 5 HV to 363 ± 20 HV, increasing by nearly four times, because the solid solution strengthening effect increased with the increase in Cr content.

(3) The compressive strength, elastic modulus and energy absorption capacity of the NiCr alloy lattice increased with the increase in Cr content and relative density. The specific strength, specific modulus and specific energy absorption of the Ni-45Cr alloy are 11.1 MP/(g/cm$^3$), 1169.1 MP/(g/cm$^3$) and 10 J/g, respectively, indicating that this alloy presents great advantages among porous materials.

**Author Contributions:** Conceptualization, P.Z. and H.Z.; methodology, P.Z. and D.H.; software, P.Z. and D.H.; validation, P.Z. and D.H.; investigation, D.H. and Y.Z.; writing—original draft preparation, P.Z.; writing—review and editing, H.Z. and Y.Z.; supervision, W.C. and H.Z.; project administration, W.C. All authors have read and agreed to the published version of the manuscript.

**Funding:** This research received no external funding.

**Institutional Review Board Statement:** Not applicable.

**Informed Consent Statement:** Not applicable.

**Data Availability Statement:** Not applicable.

**Conflicts of Interest:** The authors declare no conflict of interest.

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
