# Peer review of "Synthesis and Properties of Octet NiCr Alloy Lattices Obtained by the Pack Cementation Process"

_applsci, doi:10.3390/app13031684_

Round 1
Reviewer 1 Report
Dear Authors
The presented paper has an interesting results, please remove EDS results from page 6, correct some of SEM images with higher brightness - contrast ratio. further, when you write paper please use past tense, the research you are showing was done, also please add references related to increase of strength of the Ni - Cr system with increasing Cr content.
Thank you
Sincerely
Author Response
Response to Reviewer 1 Comments
Dear Editor and Reviewers,
Thank you for your helpful suggestions and comments. The manuscript has been carefully revised. The revisions have also been listed below in red point by point.
Point 1: The presented paper has an interesting results, please remove EDS results from page 6, correct some of SEM images with higher brightness - contrast ratio. further, when you write paper please use past tense, the research you are showing was done, also please add references related to increase of strength of the Ni - Cr system with increasing Cr content.
Response 1: Thank you for your suggestions. We have remove EDS results from page 6, correct some of SEM images with higher brightness - contrast ratio, the image below are Figure 5 in the manuscript after modification. We have changed the manuscript to the past tense and added references related to increase of strength of the Ni - Cr system with increasing Cr content.
“Dunand of Northwestern University in the United States prepared NiCr alloy foam with different Cr contents such as Ni-12Cr, Ni-22Cr and Ni-27Cr by pack-cementation process, it was found that with the increase of Cr content, the relative density of metal foam increased from 2.2% to 3.2%, and the strength increased from 0.11Mpa to 0.9MPa [13]. ”
Figure 5. EDS results of different compositions of NiCr alloy lattices: (a)(e) Ni-16Cr; (b)(f) Ni-23Cr; (c)(g) Ni-34Cr; (d)(h) Ni-45Cr, (i)(j)(k)(l) EDS mapping of Ni-45Cr lattice cross section.
We sincerely hope that the above revision is able to meet the requirement of your Journal. Please do not hesitate to contact me if there is any problem.
Best regards!
Peng Zhao

Reviewer 2 Report
The authors studied structural materials of NiCr alloys adopting a regular octahedron structure. The mechanical behavior was analyzed through microhardness and compression tests. The manuscript exhibited interesting results but there are a number of observations:
1. The manuscript must be revised, especially the English and typos;
2. P. 2, lines 49-51: The sentence in not clear;
3. In the Introduction, about the references, provide the values of the properties not only a qualitative comparison. Is not appropriate to describe the results in this section. Rewrite the last paragraph;
4. To help the readers’ understanding, explain what is the Maxwell’s criterion and pack cementation;
5. Provide more details about the furnace (as type, dimensions etc.), the microhardness and compression tests (as load, strain rate, dwell time, standard etc.);
6. In the Materials and Methods, cite which are the compositions, why they have been chosen, and how the composition was verified;
7. Most of the titles of the figures show a kind of error. Correct them;
8. In Figure 3c, the authors stated that there are no layers after the homogenization. However, what are the light and dark regions in Figure 3c?;
9. Improve the quality and size of Figure 5c,f,i,l. In Figure 5k, what is the yellow spectra?;
10. P. 6, lines 176-178: The sentence presents redundancy;
11. In Figure 8, do the values of strength and modulus correct? The legend and the values are confusing;
12. Provide more discussion about the properties among the NiCr alloys. The increase from 34 to 45 wt.%Cr didn’t improve the mechanical properties. Why?;
13. εd was not used in the equations 1.1-1.3;
14. P. 9, line 245: check if “6 times” is correct;
15. In Figure 9, the yellow column was established as specific energy absorption. Is it correct?;
16. P. 10, lines 268-270: Include the references about the studies of Zheng and Liontas. It seems that reference [32] is in the wrong place;
17. Improve the quality and size of Figure 10.
Reviewer 3 Report
Comment 1. The content of this article is almost same as of the article “Microstructure and Properties of Hollow Octet Nickel Lattice Materials” having DOI: https://doi.org/10.3390/ma15238417, published by same authors last month.
Comment 2. The abstract needs to be changed, as it just mentions the characterization techniques before synthesis and focusing the properties to attain.
Comment 3. The keywords may be written in an alphabetical order as, mechanical properties, Nickel-based lattice, …
Comment 4. In line 27, the theme of the sponge metal lattice completely changes to super alloys. How could you compare the synthesis of sponge metal lattice to that of super alloys?
Comment 5. In line 30-32, “However, in recent years, research on Ni- based lattice materials has mainly focused on pure Ni by using process of electrodeposition, and pure Ni lattice has poor oxidation resistance and corrosion resistance”.
Firstly you are saying that focus is being paid to form pure Ni-electrodeposition and at the very same time you are identifying its disadvantages!
Moreover the statement is completely wrong about Ni and evens the cited references (9 and 10) don’t even mention the corrosion and oxidation of Nickel in their paper. So I also suggest rechecking of all the references to justify your statements.
Comment 6. In line 35-37,” this can be added alloy elements to it by pack-cementation technology or chemical vapor deposition process to improve the room temperature and high temperature mechanical properties of nickel-based alloys.” Rephrase the sentence.
Comment 7. In line 40-41, “However, due to the irregular structure of the foam material with bend-dominated, the excellent mechanical properties of nickel-based alloys cannot be exerted”. This statement can’t be given without references or justification!
Comment 8. In line 44-45, “Dunand of Northwestern University in the United States”, better to just reference, no need to write any name and mention university.
Comment 9. In line 46-47“a uniform Ni-Cr-Al-Ti composition with γ/γ′ superalloy microstructure”. Give space in b/w composition and with; also re-check the manuscript for other mistakes.
Comment 10. In line 49-51, “The octahedral lattice prepared by this method has a high ultimate strength, but be-49 cause the relative density is below 2%, the actual strength was low and cannot withstand 50 large loads”, the sentence is confusing, does it achieve high strength or low strength?
Comment 11. In line 52-57, In the study of multiple lattice materials, the Octet structure has received extensive 52 attention. The Octet structure has a regular octahedron as the core with eight regular tet-53 rahedron distributing around it. Each of its nodes is connected to 12 rods, all of which are 54 the same length and diameter. Octet lattice units consist of b rods and j nodes, and con-55 form to Maxwell's criterion M= b-3j+6≥0. It is a stretch-dominated structure with extremely 56 superior performance”, almost same statement is written in paper titled “Microstructure and Properties of Hollow Octet Nickel Lattice Materials” having DOI: https://doi.org/10.3390/ma15238417, published by same authors last month.”
Comment 12. In line 62-65, The mechanical properties of the nickel-62 chromium lattice of the Octet structure have great advantages and can be used as an al-63 ternative material for load-bearing and energy absorption in high-temperature environ-64 ments.” Specify the applications, what exactly be the application for your synthesized material?
Comment 13. In line 70-71, “the polymer skeleton was pretreated and the template was immersed in an acti-70 vator solution next to deposit a palladium catalyst”, which pretreatment was carried out, is that other than immersing in activator? If yes, do mention that.
Comment 14. In line 71-74, “The sample was electroless nickel 71 plated for 10 min to deposit a metal layer with good conductivity and further electrode-72 posited to the desired thickness. The outer wall of the electrodeposited lattice was re-73 moved with sandpaper so that the polymer skeleton could be in contact with air”, completely understandable; the sample was first plated Ni electroless and electrodeposited and then again removed the surface coatings to expose the polymer skeleton to an open air?
Comment 15. In line 80-81, “the filler was poured into a high-chromium stainless steel shell and a nickel lattice 80 was inserted.” Can you add the figure of high-Cr SS shell?
Comment 16. In figure 1, better to mention each process name for better understanding.
Comment 17. In line 110, “tice sample with a rod diameter of 0.5 mm and a length of 2.5 mm.” the size is not mentioned in the figure.
Comment 18. In figure 2a, the sequence of the figures may be changed; figure c should be at left side and that of d to the right.
Comment 19. Figure 3 (a,b,c and d) is not explained properly.
Comment 20. In Figure 3 f, we have a doubt, can you please provide the JCPDS no. and reference from any other article/s to justify your statement.
Comment 21. In line 126-128, you mentioned that the NiCr lattice maintains the original structure of the hollow Ni, without any distortion and deformation in the rod of lattice, and here in Line 152-154, figure 4, you are stating the formation of larger dislocations, so which statement of yours is to be considered and in which sense?
Comment 22. Try to justify your results with other researchers to support your results. As no any reference is given in figure 6 (Microhardness).
Comment 23. In figure 7, 8 and 9, same issue, no any reference is cited to support your results.
Comment 24. In figure 7, justify your results with any other researchers; the effect of Cr in strength.
Comment 25. In line 205-209, “The density of the lattice material was the ratio of the lattice mass and the volume of 205 geometric space, the slope compression test of the initial linear strain curve was defined 206 as the elastic modulus E of the lattice material, and the non-proportional stress Rp1.0 was 207 defined as the strength σ of the lattice material. Lattice density normalization can obtain 208 lattice specific strength σ/ and specific modulus E/.” The concept is not clear. Try to explain in simple and explicit way.
Comment 26. The references should be re-checked according to the statement it justify. As there was a mistake in reference 9-10 that’s completely wrong.
Round 2
Reviewer 3 Report
Still I understand that there is no much difference, no matter the same is produced either by electrodeposition or electroless deposition. Both of results could be compared rather than writing new paper.
Can you please prepare a result Table to differentiate both papers? If satisfactory response is prepared and manuscript is updated, this may considered for publication.
Author Response
Point 1: Still I understand that there is no much difference, no matter the same is produced either by electrodeposition or electroless deposition. Both of results could be compared rather than writing new paper.
Can you please prepare a result Table to differentiate both papers? If satisfactory response is prepared and manuscript is updated, this may considered for publication.
Reply:
The first paper "Microstructure and Properties of Hollow Octet Nickel Lattice Materials "was more concerned with the effect of different length-diameter ratios of seven Octet hollow pure nickel lattice materials on deformation behavior and compression stability. And the results show that as the length-to-diameter ratio of the lattice increased, the stress–strain curve volatility of the lattice increased, and the lattice compressive deformation mode shifted from layer-by-layer and overall deformation to shear deformation in the 45° direction. The energy absorption capacity and compression stability decreased with the increase of the length-to-diameter ratio of the lattice rod because the macroporous lattice is prone to instability deformation and brittle fracture. Hollow Octet lattices with different structural parameters have essentially the same compressive strength and elastic modulus at the same density.
In the paper "Synthesis and Properties of Octet NiCr Alloy Lattice by Pack Cementation Process", we studied the second NiCr alloy lattice material that is different from the first one, and studied the effect of Cr content. And the results show that the Cr content of the NiCr alloy lattice increased from 16% to 45%, and the microhardness increased from 120±5HV to 363±20HV, increasing by nearly 4 times, because the solid solution strengthening effect will increase with the increase of Cr content. The compressive strength, elastic modulus and energy absorption capacity of the NiCr alloy lattice increased with the increase of Cr content and relative density. The specific strength, specific modulus and specific energy absorption of Ni-45Cr alloy are 11.1MP/(g/cm3), 1169.1MP/(g/cm3) and 10J/g. NiCr alloy lattices have high strength and toughness, and the strength, modulus and energy absorption capacity are advantageous in porous materials. It further improved the mechanical properties of the pure Ni lattice. It also has good oxidation resistance in theory. Therefore, it may have potential applications in high-temperature filters or heat exchangers in the future.
The material of the two articles is different, and the focus of the research is different, so it is impossible to directly compare, so we wrote a new article. A result Table to differentiate both papers are shown in the table below.

Round 3
Reviewer 3 Report
How coating thickness was controlled? Please show SEM images having coating thickness mentioned. Also, discuss the implication of it.
Also, English language need to be improved.
Author Response
Dear Editor and Reviewers,
Thank you for your helpful suggestions and comments.
Point: How coating thickness was controlled? Please show SEM images having coating thickness mentioned. Also, discuss the implication of it.
Also, English language need to be improved.
Response: Thank you for your question, we have revised this section carefully. We found that a data was confusing and we have revised it to make sure it was accurate. Lucas [1] simulated the correlation between modulus, strength and density of hollow Al2O3 lattice materials. They mentions the calculation method of the relative density and wall thickness of hollow rod lattice materials under ideal circumstances. Their research provides a feasible idea for the calculation and control of wall thickness. In this manuscript, the wall thickness of hollow NiCr alloy lattice materials is mainly affected by the electrodeposition layer and Cr deposition layer. Extending the electrodeposition time or increasing the Cr element content will increase the wall thickness. To control the wall thickness, it is necessary to further grasp the electrodeposition cathode current efficiency and the Cr element deposition efficiency during cementation. In this paper, the effect of chromium content on the mechanical properties of pure nickel lattice materials is mainly studied, and the related research on wall thickness is still ongoing.We have revised this section in the manuscript,as highlighted in page 6. Compared to the previous three groups, the large wall thickness of the lattice sample in Figure 5(d) may be caused by the angle of the sampling section view not being vertical enough.
We sincerely hope that the above revision is able to meet the requirement of your Journal. Please do not hesitate to contact me if there is any problem. We sincerely hope to receive your continuous help in future work.
Best regards!
Peng Zhao
[1]Lucas R. M.; Gregory P. P.; Carlos M. P. Reexamining the mechanical property space of three-dimensional lattice architectures. Acta Mater. 2017, 140, 424-432
